# Insight into Carbohydrate Metabolism and Signaling in Grapevine Buds during Dormancy Progression

**DOI:** 10.3390/plants11081027

**Published:** 2022-04-09

**Authors:** Valeria De Rosa, Rachele Falchi, Erica Moret, Giannina Vizzotto

**Affiliations:** Department of Agricultural, Food, Environmental and Animal Sciences, University of Udine, Via delle Scienze 206, 33100 Udine, Italy; valeria.derosa@uniud.it (V.D.R.); erica.moret@uniud.it (E.M.); giannina.vizzotto@uniud.it (G.V.)

**Keywords:** *Vitis* spp., climate change, freezing tolerance, *VvMSA*, raffinose, HPLC

## Abstract

Perennial fruit crops enter dormancy to ensure bud tissue survival during winter. However, a faster phenological advancement caused by global warming exposes bud tissue to a higher risk of spring frost damage. Tissue dehydration and soluble sugars accumulation are connected to freezing tolerance, but non-structural carbohydrates also act as metabolic substrates and signaling molecules. A deepened understanding of sugar metabolism in the context of winter freezing resistance is required to gain insight into adaptive possibilities to cope with climate changes. In this study, the soluble sugar content was measured in a cold-tolerant grapevine hybrid throughout the winter season. Moreover, the expression of drought-responsive hexose transporters *VvHT1* and *VvHT5*, raffinose synthase *VvRS* and grapevine ABA-, Stress- and Ripening protein *VvMSA* was analyzed. The general increase in sugars in December and January suggests that they can participate in protecting bud tissues against low temperatures. The modulation of *VvHT5*, *VvINV* and *VvRS* appeared consistent with the availability of the different sugar species; challenging results were obtained for *VvHT1* and *VvMSA,* suggesting interesting hypotheses about their role in the sugar–hormone crosstalk. The multifaceted role of sugars on the intricate phenomenon, which is the response of dormant buds to changing temperature, is discussed.

## 1. Introduction

Woody perennials have developed several adaptive measures to endure yearly temperature fluctuations, such as bud dormancy and cold hardiness acquisition to survive winter freezing conditions [1]. In the context of climate change, the observed increase in average surface temperatures causes an acceleration of plant phenology progression, exposing vulnerable green bud structures to a higher risk of late frost damage [2]. This is especially true for grapevine, whose development rate is very sensitive to temperature variations [3]. Both extreme winter temperatures and late freezing occurrences in spring can be highly detrimental to grapevine productivity, as observed in several areas of the world. Freezing temperatures result in a wide range of injuries in plants, spanning from the separation of cell layers and cavitation formation to the radial splitting of the trunk, tor xylem embolisms [4]. For example, winter temperatures during the 2018–2019 season in North Dakota (USA) reached a minimum of −36.8 °C, causing budbreak failure in multiple genotypes [5]. On the other hand, major economic losses were registered following the Easter freeze of April 2007 in Missouri (USA), with several wine-grape cultivars having 95% to 100% primary bud injury [6].

In several perennial species, non-structural carbohydrates have a crucial role in the budbreak process [7,8], but they are also required for basal metabolism during winter dormancy [9,10]. Moreover, soluble sugars are tightly connected to plant tolerance to cold temperatures [11,12], due to their role as osmolytes and cryoprotectants, by reducing ice nucleation within the apoplast, thus limiting freezing-induced dehydration [13]. Intracellular ice formation results not only in water subtraction but also in mechanical stress on the plasma membrane, likely lethal for cells [14]. Soluble sugars have been implicated in the stabilization of cellular membranes during dehydration stress, such as RFOs (Raffinose Family Oligosaccharides), which were implicated in liposome membranes protection during desiccation [15]. Fructans were also linked to winter hardiness improvement by direct interaction with membranes, as well as a putative movement to the apoplast during cold exposure [16,17]. In grapevine, variations in the concentration of several sugars such as sucrose, glucose, fructose, raffinose and stachyose have been associated with freezing tolerance [11,13,18]. In particular, raffinose has been shown to be mostly related to cold resistance and was shown to accumulate earlier in cold-tolerant cultivars compared to cold-sensitive ones [13]. Consistently, the induction of genes coding for raffinose synthase has been observed in grapevine flowers after a cold night, which is coherent with their role as an osmoprotectant [19]. Moreover, freezing tolerance enhancement has also been associated with bud water content reduction [20].

Despite their part in basal plant metabolism, sugars may play a crucial role, as signaling molecules, in the control of plant metabolism and development and revealing interactions that integrate environmental factors and hormone signaling [21]. Carbohydrate signaling in grapevines has been investigated mainly in the berry, an organ that accumulates high concentrations of sugars and several reviews are available on this topic [22,23], but the possible role in mediating bud dormancy progression is still largely unknown. Recently, sugar metabolism was proven as being pivotal for dormancy transition, not only as energy supply but also as a source of signal molecules [24,25,26]. As a consequence, in this complex feedback and multi-step regulation, sugar transporters and metabolic enzymes and their genetic regulation are critical.

Monosaccharides are delivered to sink tissues, such as buds, by hexose transporters (HTs). Fifty-nine putative grapevine HTs have been identified in grapevines [27], and three of these, namely VvHT1, VvHT4 and VvHT5, are located on the plasma membrane [28,29]. Genes *VvHT1* and *VvHT5* encode high-affinity H^+^-dependent glucose transporters [28,29] and were shown to be responsive to water stress, with *VvHT1* being downregulated and *VvHT5* upregulated [24]. *VvHT5* is also considered a general stress response-related gene, possibly due to its role in enhancing sink strength under stress conditions [30,31]. On the other hand, the irreversible sucrose hydrolysis by invertases in different cell compartments leads to hexoses increase, suggesting a fine-tuning of the sugar transport activity [32]. *VvHT1* was recently shown to be a target of VvMSA protein (Maturation, Stress, ABA), the only identified member of ASRs (ABA-, stress- and ripening-induced) in grapevine, hypothesized to act as a transcriptional regulator connecting sugar and ABA signaling [33,34], although many aspects of its action remain to be elucidated. ASR proteins have recently sparked interest because of their role as transcriptional regulators, suggested by their DNA-binding activity and nuclear localization [33], and as candidates for direct protein protection due to their hydrophilic nature [35]. A role in plant response to various environmental cues is strongly suggested by ASR induction following several stresses [36,37]. An interaction of VvMSA and a dehydration-responsive element-binding protein named VvDREB, mostly involved in osmotic stress and dehydration responses, was also observed in the nucleus of grape cells [38]. Taken together, this information makes *VvMSA* a good candidate for a role in integrating important physiological processes (i.e., hormones signaling, sugar accumulation, dehydration, stress response) during dormancy progression. Surprisingly, to our knowledge, no data on *VvMSA* expression and interactions in buds are available.

Wild grapevine species are typically more cold-hardy compared to cultivated *Vitis vinifera*. However, they are also more precocious in terms of budbreak timing, which paradoxically puts them at a higher risk of spring frost damage, with differences depending on cultivar [39]. In fact, for putative evolutionary reasons, deacclimation was observed to proceed much faster in wild species such as *Vitis riparia* and *Vitis amurensis*, routinely used by breeders to introduce resistant phenotypes in *V. vinifera*. Therefore, a better understanding of dormancy physiology and regulation also for these species is desirable [40].

This work aims to explore soluble sugars metabolism in the context of low-temperature-induced responses during endo- and ecodormancy in buds of a hybrid of renowned cultivar Merlot, UD 31-103 (Merlot × Kozma 20-3). This hybrid is a disease-resistant selection of the University of Udine, and it was chosen for its improved resistance to low temperatures due to the presence of a cold-tolerant grape cultivar (*Vitis amurensis*) in its pedigree. This investigation will provide new insights to deepen our understanding of the regulation of dormancy, paving the way for effective frost mitigation strategies.

## 2. Materials and Methods

### 2.1. Plant Material

The red grapevine interspecific hybrid UD 31-103 (Merlot × Kozma 20-3), tolerant to minima of −20 °C, was field-grown at the Experimental Farm “A. Servadei” (46°04′ N 13°14′ E, University of Udine, Northern Italy). During the 2019–2020 winter season, buds were regularly collected at 9–10 AM every ~15 days from October to March, in proximity to budbreak. Buds were stored at −80 °C for gene expression analysis and soluble sugars measurements.

### 2.2. Soluble Sugars Extraction

Soluble sugar extraction was performed based on a previously tested protocol [18]. Three biological replicates of 10 buds were ground in liquid nitrogen and subsequently freeze-dried for 72 h. Forty ± 5 mg of ground sample powder were moved to 2 mL tubes, and 1 mL of 75% ethanol (*v*/*v*) at room temperatures was added for incubation of 3 h. Samples were continuously shaken during incubation and vortexed at maximum speed for 1 min every 30 min. After 5′ centrifugation at 6700× *g*, supernatants were collected and dried in a centrifugal vacuum concentrator. All steps were repeated twice for each sample.

### 2.3. HPLC Analysis

Soluble sugars were separated using a 250 mm long Ultra Amino column (Restek S.r.l., Cernusco sul Naviglio, Italy) with a 4.6 mm internal diameter and 5 µm particle size, equipped on a 1260 Infinity HPLC system (Agilent Technologies, Santa Clara, CA, USA) with autosampler, quaternary pump and refractive index detector. An acetonitrile/water mixture (70/30) was used as the mobile phase (1 mL·min^−1^). The oven and detector were set at 30 °C. Before injection, dry soluble sugars extracts were added to 300 µL of the mobile phase and thoroughly mixed using a vortex for 30″. To ensure complete sample solubilization, sonication in an ultrasound bath was performed for 5′. Finally, 50 µL of the sample was injected following filtration.

Standard solutions of glucose, fructose, sucrose and raffinose (Sigma-Aldrich, St. Louis, MO, USA) were used for sugar detection and quantification. Calibration curves were constructed injecting each sugar standard at concentration ranges: 25,000–25 μg/mL glucose, 20,000–25 μg/mL fructose, 25,000–25 μg/mL sucrose, and 1990–7.5 μg/mL raffinose. Sugar quantification was calculated from the peak area using Agilent OpenLab CDS ChemStation Edition (Version C.01.03) software (Agilent Technologies, Santa Clara, CA, USA).

### 2.4. Gene Expression Analysis

For each sampling time, RNA extraction was performed from 3 biological replicates of 10 buds using the Spectrum™ Plant Total RNA kit (Sigma-Aldrich, St. Louis, MO, USA). cDNA was synthesized with QuantiTect^®^ Reverse Transcription kit (Qiagen, Hilden, Germany), and qPCR was carried out with SsoFast™ EvaGreen^®^ Supermix (Bio-Rad, Hercules, CA, USA) as previously described [41]. Primers used to detect gene expression are listed in Appendix A. Ubiquitin Conjugating Factor (CF203457; VIT_19s0015g01190) was selected as a housekeeping gene for qPCR data normalization.

### 2.5. Statistical Analysis

All statistical analyses were performed with SigmaPlot 14.0 (https://systatsoftware.com/) using one-way ANOVA and Tukey HSD as post hoc tests for all pairwise comparison procedures. Statistical analysis of sugar content data was performed separately for each sugar.

## 3. Results

### 3.1. Soluble Sugars Accumulation Dynamics in Dormant Buds

Hexoses and sucrose content were successfully detected and quantified by HPLC analysis in buds of the hybrid UD 31-103 (Figure 1a) throughout the 2019–2020 winter season. Additionally, raffinose concentrations were measured, although it appeared to be stably less concentrated, in all samples, as compared to sucrose and hexoses (Figure 1b).

Changes in the concentration of hexoses and sucrose followed the same tendency. According to the obtained results, all sugars sharply peaked on 15 December, reaching seasonal maximum levels. Then, sugar levels were observed to gradually decrease up to the second half of January. A second peak, smaller than the first, was detected on 11 February.

Sucrose was the most abundant sugar in buds, while monosaccharides differed from each other for the concentration that was lower for glucose. Raffinose concentration was the lowest and appeared more modulated; in fact, it showed a transient increase in early November, followed by the highest concentration on 15 December, similarly to all the other sugar species. Raffinose content then decreased until 9 January and remained stable for the entire month, when, at the end of the winter, the concentration of the trisaccharide became almost undetectable.

### 3.2. Seasonal Variation of Temperature

Daily temperature data, recorded by the S. Osvaldo (Udine, Italy) weather station and managed by ARPA FVG, are represented in Figure 2 and detailed in Appendix A. The mean daily temperatures in the 2019–2020 winter season regularly decreased starting from November, reaching their minimum values in the last days of December and the first week of January, although temperatures around 3 °C were observed since the beginning of December. Interestingly, during the coldest winter period, a transient increase in temperature was observed between 15 and 23 December, when the environmental temperature averaged about 10 °C. After that, the above-mentioned minimum was reached and then slowly progressed towards increasing temperatures in February and March.

### 3.3. Quantification of Gene Expression in Buds during the Winter Season

The transcriptional regulation of five selected genes was investigated by qRT-PCR in order to possibly explain the sugars accumulation patterns and substantiate their role as osmoprotectants and signaling molecules (Figure 3). In detail, hexose transporters *VvHT1* and *VvHT5* were selected due to their involvement in different stress responses, in tight connection with VvMSA protein, the only identified member of ASRs in grapevine able to integrate sugar, stress and hormonal signaling. In addition, the expression of *VvINV* [42] and *VvRS* was determined because of their pivotal role in the production of hexoses and raffinose, respectively.

The gene encoding the hexose transporter VvHT5 showed a clear expression pattern characterized by an upregulation in the central phase of the considered period. Specifically, a peak on 9 January and a decrease to a new significant minimum in late February and March were detected.

On the contrary, *VvHT1* transcription exhibited fairly complementary dynamics compared to *VvHT5*. In detail, *VvHT1* expression displayed its minimum from November to January and seemed tendentially upregulated from the beginning of February onwards. In addition, cell wall invertase *VvINV* expression appeared modulated in a very complex way. Notably, downregulation was observed from December to January–February, while the highest *VvINV* transcription was detected in February and March.

*VvMSA* expression [43] was successfully detected in grapevine buds of UD 31-103. Results show that *VvMSA* was significantly and sharply upregulated in spring. In detail, although higher expression levels were measured in October as compared to the following time-points, a general downregulation was observed during the fall–winter season, but a sharp and significant increase in gene expression was detected on 10 March.

Moreover, the raffinose synthase encoding gene (*VvRS*) was monitored in buds of UD 31-103 throughout the 2019–2020 winter season. Despite the low levels of significance, *VvRS* displayed a general higher expression until the beginning of January and a tendential downregulation in the following time-points, reaching a minimum in February and March, in accordance with raffinose dynamics.

## 4. Discussion

Climate change is not only registered as torrid summers but also warmer winters or acute cold weather episodes [2]. Recently, dormancy regulation has gained additional interest due to global warming and unpredictable temperature fluctuations. Cold hardiness kinetics and budbreak phenology are strictly connected and affected by a dormancy state [44]. In fact, early cold hardiness loss often implies higher frequencies of late frost damage in sensitive species [45]. Grapevine cultivars have been shown to be differentially sensitive to warm spells during winter; however, sustained cold winters are necessary for high cold hardiness levels to be maintained [46]. Soluble sugars accumulation during cold acclimation play a well-documented role in freezing tolerance acquisition [26]. However, sugars are also known as signaling molecules involved in the combination of internal and external stimuli in different developmental stages and stress responses [47,48], as well as a source of energy for primary metabolism [8]. The role of sugar metabolism in bud dormancy transitions is still largely unknown. A better understanding of the mechanisms underpinning dormancy progression and the key players in this process could be helpful in increasing the sustainability of grapevine cultivation as climate variation increases.

Cold hardiness of buds collected from the UD 31-103 hybrid during the 2019–2020 winter season has recently been documented, showing maximum levels in December and in the first part of January [49].

As a rule, dormant buds are considered to be in an inactive state with reduced metabolic activities; therefore, the dormancy progression and the transition toward budbreak certainly involve the general reprogramming of carbohydrate metabolism [26]. Although it may be difficult to completely understand the data collected in this preliminary experiment and to establish their cause–effect relationship, the work might help in defining the complex pattern of metabolic reactions occurring within dormant buds.

Indeed, significant changes in sugars and related genes transcription were observed. The raffinose accumulation dynamics appear to confirm its greater correlation to freezing tolerance compared to other soluble sugars [19], with the increase in raffinose already detectable in the early stages of the considered period. The transient decline on 25 November suggests that raffinose can rapidly respond to changes in temperature, as the highest maximum of the period was recorded on that date (Figure 2). Consistently, *VvRS* expression almost mirrors this trend, and its downregulation from the beginning of February corresponds to the increase in average daily temperatures during the 2019–2020 winter season following the coldest interval (Figure 2). To further corroborate this *VvRS* responsiveness to rapid temperature changes, a simple relation can be drawn between expression in early January and the coldest temperatures of the season (Figure 2). *VvRS* expression pattern, apparently inconsistent with raffinose levels at this stage, highlights the complexity of its metabolism [50] that requires further attention. The presence of multiple *VvRS* homologs in the grapevine genome also supports the possibility of differential functions or timings of activation [51]. The induction of the same gene coding for raffinose synthase was described in inflorescence exposed to cold, allowing us to hypothesize a role for raffinose as an osmoprotectant [19]. However, it is possible to hypothesize that the accumulation of sugars, probably connected with the acquisition of cold hardiness, can be driven by many factors, of which the external temperature represents only a part. For this reason, the environmental temperature (Figure 2) can only in part explain buds’ carbohydrate content, which, on the contrary, appears reasonably related to low temperatures tolerance (measured as DTA, Figure 4).

As expected, the accumulation pattern of sucrose and hexoses appears more difficult to be directly related to low temperatures occurrences. This is specifically due to the difficulties in discriminating the role of sugars as signaling molecules from their contribution as metabolic substrates and/or osmolytes [52]. However, soluble sugars levels were low in buds at the beginning of dormancy and then increased gradually with a major peak in December and a following smaller increase during February. These results are fairly in agreement with previous findings in grapevine and other species, suggesting that sucrose and hexoses can also contribute to improving the freezing tolerance by lowering the freezing point of free water and inhibiting the formation of ice crystals [26,53]. However, the relative amounts of sugars with fructose higher than glucose might reflect the utilization of both glucose and sucrose in raffinose biosynthesis. This speculation further corroborates the difficulty in mechanistically explaining the relative proportion of the different sugars measured, given their diversified function and their metabolic crosstalk.

On the other hand, the expression of hexose transporters encoding genes appeared only partially coherent with the accumulation pattern of fructose and glucose. In detail, *VvHT1* is nearly downregulated during winter months, while *VvHT5* is significantly upregulated in a complementary way around the same time period. These results seem consistent with previously reported drought-induced *VvHT5* upregulation and *VvHT1* downregulation [30]. In addition to this, recent evidence documented the upregulation of an unspecified hexose transporter in cold-treated *Vitis amurensis* seedlings while showing no obvious difference in *V. vinifera* [54]. Given the similarities of plants’ adaptation mechanisms to cold and drought stress [55,56], our results allow suggesting that hexoses transporters encoding genes examined in this work display comparable behavior as observed in drought-stress responses. In particular, *VvHT5* upregulation could be related to freezing-induced dehydration stress, leading to hexoses uptake and enhanced cold tolerance. In addition, a downregulating role of glucose on *VvHT1* expression and glucose uptake has been recognized [57], and the pattern identified in our work should be consistent with this interaction, being the accumulation profile of glucose almost complementary to that of *VvHT1* transcripts. Interestingly, *VvHT1* expression is controlled by VvMSA during berry development [58]. To our knowledge, the results presented in this study are the first example of documented *VvMSA* expression in grapevine buds. The expression pattern suggests a role for VvMSA in the context of dormancy release, and its double regulation by both ABA and sugars makes it an interesting target for future investigations as a relevant key player in dormancy progression. Although an immediate connection between soluble sugars and *VvMSA* expression in grapevine buds cannot be stated from these data, as seen in other grapevine tissues [33,34], the coordination between *VvHT1* and *VvMSA* expression supports the hypothesis of a possible interplay between the two genes also in buds.

Cell wall invertases are generally regarded as providing substrates for metabolism, but in many conditions, the glucose and fructose produced are mainly used in osmoregulation [59]. Recent evidence collected in tomatoes has indicated that cell wall invertases play an important role in chilling tolerance by regulating sugar content [60]. Ice nucleation events in plant tissues take place outside living cells, namely in the apoplast, from where freezing can spread to the symplast [61]. Cell wall invertases are located in the apoplast, where they hydrolyze sucrose into glucose and fructose, doubling their osmoprotective function; the evidence collected in this study allows us to hypothesize that generated hexoses could participate in establishing freezing tolerance in grapevine buds. However, our results do not point out a correlation between hexoses amount and *VvINV* expression. This is not surprising since cell wall invertases can undergo post-transcriptional regulation and can be inactivated by binding to inhibitor proteins [59]. In any case, cell wall invertases have a recognized role in regulating developmental transitions [62], and their expression was induced shortly before bud break in buds of peach [63], suggesting the catabolic activity can increase the sink strength, which, in turn, triggers a commitment to bud burst [64]. This hypothesis appears consistent with our findings, being *VvINV* significantly upregulated in February and March when hexoses also showed a slight increase. This upregulation, concomitant to the gradual rise of temperatures starting in February (Figure 2), appears in accordance with the proximate growth restart of buds.

## 5. Conclusions

In conclusion, this work presents evidence of multiple sugar-related responses taking place inside buds of a cold-tolerant *Vitis* hybrid. Although not conclusively, it is possible to attribute a specific role to raffinose in bud acclimation and tolerance to low temperatures. The accumulation pattern of sucrose and hexose is more intricate, as they are simultaneously metabolic substrates and osmoprotectants. However, the increase in these sugars in December and January suggests that they can participate in protecting bud tissues against low temperatures. In this context, it has been possible to relate the modulation of *VvHT5*, *VvINV* and *VvRS* to the availability of the different sugar species. Conversely, challenging results were obtained for *VvHT1* and *VvMSA* genes (Figure 4). However, VvMSA has been proposed as a point of convergence in the sugar–hormone crosstalk [65] and as a key regulator of *VvHT1.* The first evidence on *VvMSA* expression in grapevine buds and its coordination with *VvHT1* make them interesting candidates for a possible role in bud phenological advancement towards budbreak.

New insights have been added to the full understanding of the complex phenomenon of dormancy progression. An integrative approach is still required in order to untangle the key molecular pathways involved and to investigate how environmental conditions affect this phenological transition.

## Figures and Tables

**Figure 1 plants-11-01027-f001:**
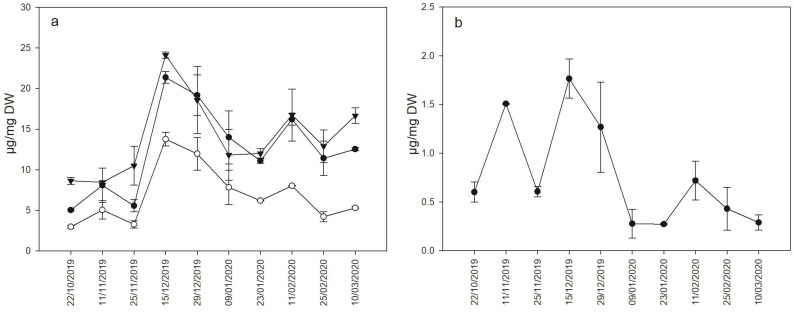
(**a**) Accumulation dynamics of fructose, glucose and sucrose in buds of UD 31-103 throughout the 2019–2020 winter season. In (**b**), the raffinose accumulation pattern in the same samples is reported. Results are expressed as the mean of 3 biological replicates of 10 buds ± standard error. Statistical analysis is provided in Appendix A.

**Figure 2 plants-11-01027-f002:**
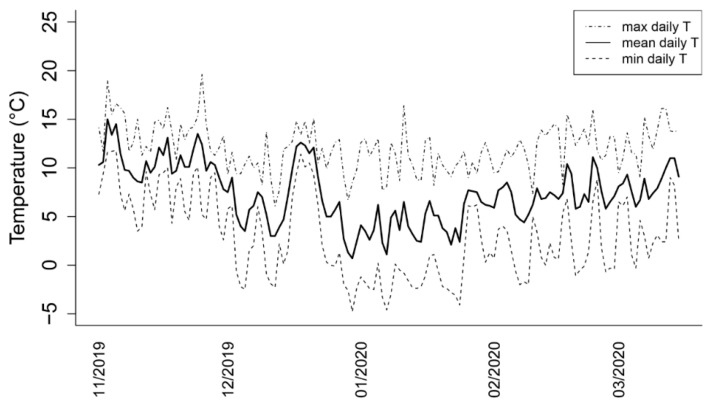
Daily temperatures data registered during the 2019–2020 winter season.

**Figure 3 plants-11-01027-f003:**
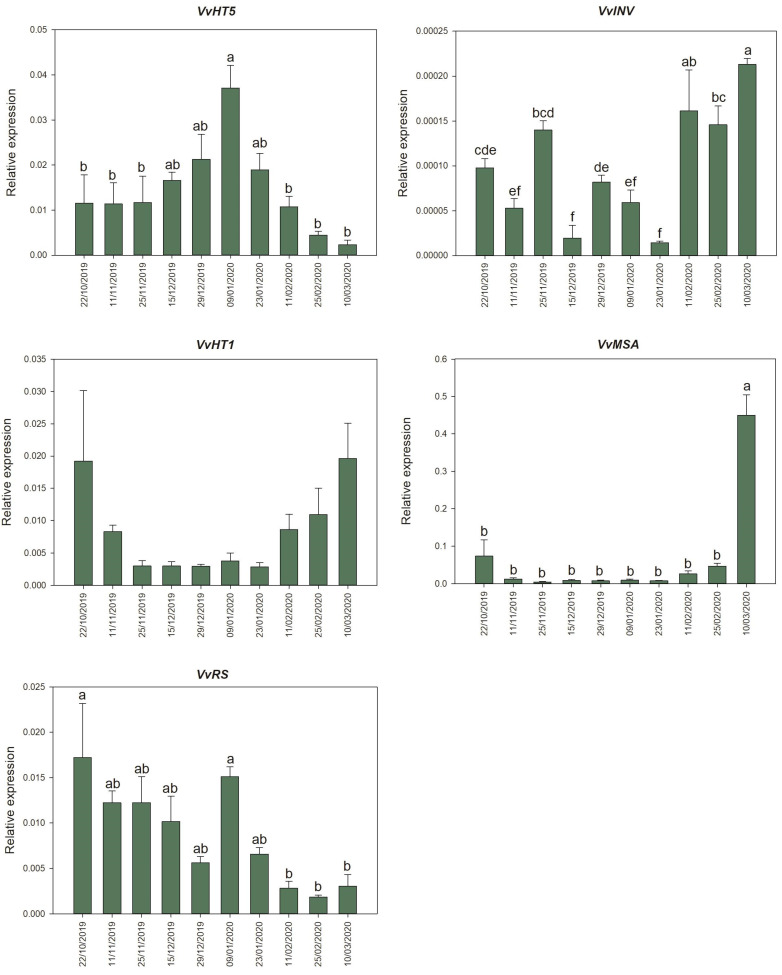
qPCR analysis of buds of hybrids UD 31-103 throughout the 2019/2020 winter season. Results are expressed as the mean of 3 biological replicates of 10 buds ± standard error. *VvHT5* = Hexose Transporter 5; *VvHT1* = Hexose Transporter 1; *VvINV* = Cell Wall Invertase; *VvMSA* = Maturation, Stress, ABA; *VvRS* = raffinose synthase. Letters represent the results of statistical analyses; in the *VvHT1* expression pattern, no significant difference was found.

**Figure 4 plants-11-01027-f004:**
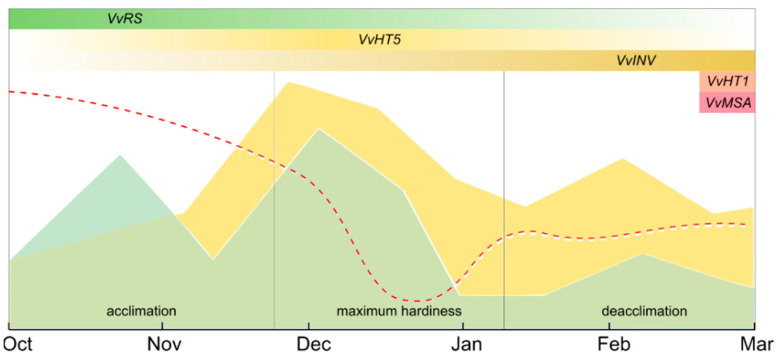
Relationship between sugars accumulation patterns (raffinose in green, hexoses in yellow) and dormancy progression in the grapevine interspecific hybrids UD 31-103. Dashed red line approximately depicts the gain and loss of cold-hardiness during the dormant season (data from [49]). Potential regulatory roles of genes at different stages are shown in boxes. The trends are purely indicative and irrespective of the quantitative relationships among the various parameters.

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
