# Peer review of "Insight into Carbohydrate Metabolism and Signaling in Grapevine Buds during Dormancy Progression"

_plants, 2022, doi:10.3390/plants11081027_

Round 1

Reviewer 1 Report

the monitoring of sugars and raffinose over the course of the winter period is interesting and a selling point of the manuscript. However, I feel authors should better connect the expression of the chosen hexose transporter (HT) genes to the sugar accumulation. It is a bit unclear why these genes have been chosen. After all these HT genes encode plasma membrane sugar transporters and it has been shown that grapevine and other plants accumulate sugars during cold periods mainly in vacuoles. So I would have suggested to the authors to at least test expression of the tonoplast located TST genes. They already have the cDNA and material, so this seems manageable and would increase impact of the manuscript.

Author Response

Cold/freeze responses at the cellular level are characterized by a general reprogramming of gene expression and metabolic fluxes.

One of the most prominent biochemical alterations during cold acclimation in buds is the accumulation of soluble sugars (acting as osmolytes but also in stabilizing biological membranes). Sugars can contribute in improving the freezing tolerance by lowering the freezing point of free water and inhibiting the formation of ice crystals [20,45]. However, controlling cellular osmotic balance and ion homeostasis, i.e., intracellular transportation of water, transportation of toxic ions inside the vacuole, synthesizing osmolytes in the cytoplasm, heat shock proteins, and activation of the enzymatic as well as non-enzymatic antioxidant systems are the most important conserved mechanism that confer stress tolerance in plants (https://doi.org/10.1080/15592324.2021.1913306). Therefore, sugars can contribute to the acclimation process, but it is well-known that several key-players participate to the process and sugars are also involved in several important metabolic or signaling pathways. 

Taken together, this information led us to think that in order to avoid the risk of extracellular/intracellular freezing, the key point is the accumulation of sugars in the cytoplasm or in the apoplast of the bud cells. The compartmentation in the vacuole is obviously not to be excluded, but it did not seem functional to our discussion.

As mentioned in the Introduction, few genes were selected for transcriptional analyses. In detail, VvHT5 and VvHT1 are responsive to dehydration stress and VvHT1 was recently shown to be a target of VvMSA protein [22-26]. They are certainly not all the genes involved in the transport/metabolism of sugars, but we believe that even adding the TST expression profiles, the possible conclusions would be controversial and purely speculative. On the other hand, the debatable role of sugar transporters is confirmed by the contrasting expression profiles of the two VvHTs, detected in this work.

It is clear that only much more detailed and advanced analyses could allow us to define a cause-effect relationship between the transcription of a transporter, the associated sugar compartmentation and its physiological role. We do not claim to give an exhaustive explanation of the role of all metabolic species in grapevine buds. To our knowledge, little is known about the evolution of the sugar patterns in grapevine buds and here we described these patterns trying to provide a likely explanation to them, keeping in mind the complex crosstalk of stress responses.

Reviewer 2 Report

The manuscript by Valeria De Rosa et. al represents the data on the alteration in the content of four soluble sugars (glucose, fructose, sucrose, and raffinose) and the expression of five genes (four genes related to sugar metabolism and one stress-related gene) in cold-tolerant grapevine bud tissue under field conditions during period from the end of October till March. Authors suggest that soluble sugars are not only osmoprotectants and cryoprotectants, but also signaling molecules regulating the expression of stress-related genes in grapevine buds. The studied biological object is interesting and important, and the hypothesis is very interesting. Unfortunately, the experimental evidence presented is insufficient to draw the conclusion. There is a clear need for additional evidence, based either on experiments with exogenous grapevine treatment with mentioned sugars, to confirm their ability to regulate the expression of stress-related genes, or a broader picture of metabolic changes and gene expression in overwintering grape buds has to be presented to prove the specificity of the observed effect. In light of the above, it is advisable to obtain more experimental data supporting the hypothesis.   

Minor comments:

In the abstract, the abbreviations of genes are present without explaining what those genes are: “the expression of drought-responsive VvHT1 and VvHT5, VvRS and VvMSA was analyzed”. Moreover, this sentence is followed by a sentence about sugars (“The general increase of these sugars in December and January suggests that they can participate in protecting bud tissues against low temperatures”), which adds uncertainty. 

Please, clarify, what do numbers mean: 25000 – 25 μg/mL glucose, 20000 – 25 μg/mL fructose, 25000 – 119 25 μg/mL sucrose, and 1990 – 7.5 μg/mL raffinose.

No statistical differences are indicated on the graphs of Figure 1.

The indications on graphs and figure legends do not coincide. “Figure 1. (a) Accumulation dynamics of total hexoses (•) and sucrose (°)”, but on the graphs sucrose, hexose and glucose are indicated.

Drastic changes in VvINV expression between the end of January and the beginning of February are observed, but this observation is not discussed. 

It is written: “VvRS displayed a general higher expression until the beginning of January and a tendential downregulation in the following time-points, reaching a minimum in February and March, in accordance with raffinose dynamics”. But indeed, according to the graphs, the maximum of VvRS expression in January corresponds to the minimum in raffinose level. This also should be discussed.

It is not indicated that samples were collected at the same time of the day, but circadian rhythm may influence sugar content. 

Fig. 3 Please, correct “Dic”

Author Response

We agree with the Reviewer and we would be very happy to provide here the conclusive proof of our hypotheses.

However, it will be agreed that, the replication of one or more trials, for an experiment carried out on tree plants, grown in the open field, is prone to seasonality and environmental variations. Even assuming experiments under controlled conditions, the results, undoubtedly more complete and appealing, would be available in one or two years.

All minor comments were addressed in the manuscript.

Please see attachments.

Reviewer 3 Report

Revision to

Title: Insight into carbohydrate metabolism and signaling in grape vine buds during dormancy progression

Authors: Valeria De Rosa, Rachele Falchi, Erica Moret, Giannina Vizzotto

Abstract Journal: Plants

Manuscript number: plants-1653275

General remarks: The manuscript by De Rosa et al. reported an investigation on a grapevine hybrid genotype subjected to winter season (October-March) to study the dormancy progression. The authors mainly focused on soluble sugar contents of buds. The final purpose of the authors is to study the soluble sugars metabolism in the context of low temperature-induced responses during endo- and eco-dormancy in buds. The manuscript is very interesting but in my opinion the authors made a conceptual mistake. Their compared their data with date, showing both sugars and expression analysis according to the different considered periods of the year and not with temperatures. I suggest to the authors to use the supplemental data about temperatures in the main figures. Then results should better detailed based on the comparison between results and temperatures. Another lacking point is in the discussion. This section is not written in depth. Authors should better argued their data clarifying the effective significance (certainly of value), improving the comparison of temperature stress and differences of sugar content and expression analysis. Finally a moderate revision of the English fluency is necessary.

Based on these considerations this manuscript will be suitable for publication on Plants after major revisions.

Introduction:

The intro is too much generic in some parts, and should be better developed, detailing the importance of the temperature stress for grapevine (e.g. cite some regions, specify the detrimental effects of temperature stress).

Line 34-40: the concept of sugar roles counteracting temperatures stresses should be more in details.

Please mention the hybrid genotype in introduction, motivating the reasons that made it the model organism for the authors.

Methods:

Please mention the geographic coordinates of the experimental farm.

Please add an additional table showing mean temperatures in the experimental period.

Lines 99-105: Authors should improve the description of soluble content extraction method.

Lines 125-140: Please mention the housekeeping gene used for qRT-PCR and cite reference of this.

Please improve a paragraph in methods section describing statistical methods. This way you can avoid detailing stats in figure captions

Results and Discussion

See general remarks

In my opinion authors should not cite Figure 3 only in conclusion. Authors should describe this figure or in the results section or should argued this in the discussion section.

Figures

Authors should move table 1 in supplementary.

Please improve the figures quality. In some case are difficult to see

Author Response

We revised results and discussion trying to give greater importance to the trend of the external temperature and, according to reviewer’s suggestion, we moved Figure S1 to the main text. However, we consider the sugars content in buds (strictly related to dormancy progression and cold hardiness acquisition) as a result of several intricate mechanisms in which external temperature represents only a part.

We modified the manuscript according to all the other appropriate Reviewer’s comments. Please see attachment.

Round 2

Reviewer 2 Report

One can agree with the authors' statement that the repetition of the presented field experiment will require an additional one or two years, which may be accompanied by completely different weather conditions. This circumstance should be taken into account. Since additional experimental evidence cannot be obtained, and given the novelty and practical significance of the research, as well as the fact that the authors present their point of view only as a "hypothesis", the work can be considered as meeting the requirements for publication. The text of the manuscript has been improved, and all previous comments have been taken into account. The presented study fits very well to the topic of the collection "New Trends in Plant Science in Italy".

Minor comment: Please, write what means “ASR” and “VvMSA” in the abstract.

Reviewer 3 Report

The v2 version of the manuscript by De Rosa et al. is significantly improved. In my opinion is now suitable for pubblication on Plants.